# Reflected Schrödinger Bridge for Constrained Generative Modeling

**Wei Deng**[1]   **Yu Chen**[1]   **Nicole Tianjiao Yang**[2]   **Hengrong Du**[2]   **Qi Feng**[2]   **Ricky T. Q. Chen**[5]

[1]Machine Learning Research, Morgan Stanley
[2]Department of Mathematics, {Emory, Vanderbilt, Florida State} University
[5]Meta AI (FAIR)

## Abstract

Diffusion models have become the go-to method for large-scale generative models in real-world applications. These applications often involve data distributions confined within bounded domains, typically requiring ad-hoc thresholding techniques for boundary enforcement. Reflected diffusion models [Lou and Ermon, 2023] aim to enhance generalizability by generating the data distribution through a backward process governed by reflected Brownian motion. However, reflected diffusion models may not easily adapt to diverse domains without the derivation of proper diffeomorphic mappings and do not guarantee optimal transport properties. To overcome these limitations, we introduce the Reflected Schrödinger Bridge algorithm—an entropy-regularized optimal transport approach tailored for generating data within diverse bounded domains. We derive elegant reflected forward-backward stochastic differential equations with Neumann and Robin boundary conditions, extend divergence-based likelihood training to bounded domains, and explore natural connections to entropic optimal transport for the study of approximate linear convergence—a valuable insight for practical training. Our algorithm yields robust generative modeling in diverse domains, and its scalability is demonstrated in real-world constrained generative modeling through standard image benchmarks.

## 1 INTRODUCTION

Iterative refinement is key to the unprecedented success of diffusion models. They exhibit statistical efficiency [Koehler et al., 2023] and reduced dimensionality dependence [Vono et al., 2022], driving innovation in image, audio, video, and molecule synthesis [Dhariwal and Nichol, 2022, Ho et al., 2022, Hoogeboom et al., 2022, Bunne et al., 2023]. However, diffusion models do not inherently guarantee optimal transport properties [Lavenant and Santambrogio, 2022] and often result in slow inference [Ho et al., 2020, Salimans and Ho, 2022, Lu et al., 2022]. Furthermore, the consistent reliance on Gaussian priors imposes limitations on the application potential and sacrifices the efficiency when the data distribution significantly deviates from the Gaussian prior.

The predominant method for fast inference originates from the field of optimal transport (OT). Notably, the (static) iterative proportional fitting (IPF) algorithm [Kullback, 1968, Ruschendorf, 1995] addresses this challenge by employing alternating projections onto each marginal distribution. This algorithm has showcased impressive performance in low-dimensional contexts [Chen and Georgiou, 2016, Pavon et al., 2021, Caluya and Halder, 2022]. In contrast, the Schrödinger bridge (SB) problem [Léonard, 2014] introduces a principled framework for the dynamic treatment of entropy-regularized optimal transport (EOT) [Villani, 2003, Peyré and Cuturi, 2019]. Recent advances [De Bortoli et al., 2021, Chen et al., 2022b] have pushed the frontier of IPFs to (ultra-)high-dimensional generative models using deep neural networks (DNNs) and have generated straighter trajectories; Additionally, SBs based on Gaussian process [Vargas et al., 2021] demonstrates great promise in robustness and scalability; Bridge matching methods [Shi et al., 2023, Peluchetti, 2023] also offers promising alternatives for solving complex dynamic SB problems.

Real-world data, such as pixel values in images, often exhibits bounded support. To address this challenge, a common practice involves the use of thresholding techniques [Ho et al., 2020] to guide the sampling process towards the intended domain of simple structures. Lou and Ermon [2023] introduced reflected diffusion models that employ reflected Brownian motion on constrained domains such as hypercubes and simplex. However, constrained domains

---

Deng, Chen, Yang, and Du contributed equally to this work.
Correspondence: weideng056@gmail.com

*Accepted for the 40th Conference on Uncertainty in Artificial Intelligence* (UAI 2024).

on general Euclidean space with optimal transport guarantee are still not well developed. Moreover, Lou and Ermon [2023] relies on a uniform prior based on variance-exploding (VE) SDE to derive closed-form scores, and the popular variance-preserving (VP) SDE is not fully exploited.

To bridge this gap, we propose the *Reflected Schrödinger Bridge* (SB) to model the transport between any smooth distributions with bounded support. We derive novel reflected forward-backward stochastic differential equations (reflected FB-SDEs) with Neumann and Robin boundary conditions and extend the divergence-based likelihood training to ensure its confinement within any smooth boundaries. We further establish connections between reflected FB-SDEs and EOT on bounded domains, where the latter facilitates the theoretical understanding by analyzing the convergence of the dual, potentials, and couplings on bounded domains. Notably, our analysis provides the first non-geometric approach to study the uniform-in-time stability w.r.t. the marginals and is noteworthy in its own right. We empirically validate our algorithm on 2D examples and standard image benchmarks, showcasing its promising performance in generative modeling over constrained domains. The flexible choices on the priors allow us to choose freely between VP-SDE and VE-SDE.

## 2 RELATED WORKS

**Constrained Sampling** Bubeck et al. [2018] studied the convergence of Langevin Monte Carlo within bounded domains. His work revealed a polynomial sample time for log-concave distributions, which is later extended to non-convex settings by Lamperski [2021]. Furthermore, the exploration of constrained sampling in challenging scenarios with ill-conditioned and non-smooth distributions was explored by Kook et al. [2022], who leveraged Hamiltonian Monte Carlo techniques. Other constrained sampling works include proximal Langevin dynamics [Brosse et al., 2017] and mirrored Langevin dynamics [Hsieh et al., 2018].

**Constrained Generation** De Bortoli et al. [2022], Huang et al. [2022] studied the extension of diffusion models on Riemannian manifolds, and the convergence is further analyzed by De Bortoli [2022]. This groundwork subsequently motivated follow-up research, including implicit score-matching loss via log-barrier methods and reflected Brownian motion [Fishman et al., 2023] and Schrödinger bridge [Thornton et al., 2022] on the Riemannian manifold. Alternatively, drawing inspiration from the popular thresholding technique in real-world diffusion applications, Lou and Ermon [2023] proposed to train explicit score-matching loss based on reflected Brownian motion, which demonstrated compelling empirical performance. Mirror diffusion models [Liu et al., 2023a] studied constrained generation on convex sets and found interesting applications

in watermarked generations. Liu et al. [2023b] employed Doob's h-transform to learn diffusion bridges on various constrained domains. The study of reflected Schrödinger bridge was initiated by Caluya and Halder [2021] in the control community and has shown remarkable performance in low-dimensional problems.

## 3 PRELIMINARIES

Diffusion models [Song et al., 2021b] have achieved tremendous progress in (text-to-)image generation. However, real-world data (such as the bounded pixel space in images) often comes with bounded support. As such, practitioners often employ ad-hoc thresholding techniques to project the data to the desired space, which inevitably affects the theoretical understanding and hinders future updates.

To generalize these techniques, Lou and Ermon [2023] utilized reflected Brownian motion to train explicit score-matching loss in bounded domains. They first perturb the data with a sequence of noise and then propose to generate the constrained data distribution via the corresponding reflected backward process [Williams, 1987, Cattiaux, 1988].

$$d\mathbf{x}_t = \boldsymbol{f}(\mathbf{x}_t, t)dt + g(t)d\mathbf{w}_t + d\mathbf{L}_t, \quad \mathbf{x}_0 \sim p_{\text{data}} \subset \Omega \tag{1a}$$

$$d\mathbf{x}_t = \left[\boldsymbol{f}(\mathbf{x}_t, t) - g(t)^2 \nabla \log p_t(\mathbf{x}_t)\right] dt \tag{1b}$$
$$+ g(t)d\overline{\mathbf{w}}_t + d\overline{\mathbf{L}}_t, \quad \mathbf{x}_T \sim p_{\text{prior}} \subset \Omega$$

where $\Omega$ is the state space in $\mathbb{R}^d$; $\boldsymbol{f}(\mathbf{x}_t, t)$ and $g(t)$ are the vector field and the diffusion term, respectively; $\mathbf{w}_t$ is the Brownian motion; $\overline{\mathbf{w}}_t$ is another independent Brownian motion from time $T$ to 0; $\mathbf{L}_t$ and $\overline{\mathbf{L}}_t$ are the local time to confine the particle within the domain and are defined in Eq.(19); the marginal density at time $t$ for the forward process (1a) is denoted by $p_t$. $\nabla \log p_t(\cdot)$ is the score function at time $t$, which is often approximated by a neural network $s_\theta(\cdot, t)$. Given proper score approximations, the data distribution $p_{\text{data}}$ can be generated from the backward process (1b).

## 4 REFLECTED SCHRÖDINGER BRIDGE

Although reflected diffusion models have demonstrated empirical success in image applications on hypercubes, extensions to general domains with optimal-transport guarantee remain limited [Lavenant and Santambrogio, 2022]. Notably, the forward process (1a) requires a long time $T$ to approach the prior distribution, which inevitably leads to a slow inference [De Bortoli et al., 2021]. To solve that problem, the dynamic SB problem on a bounded domain $\Omega$ proposes to solve

$$\inf_{\mathbb{P} \in \mathcal{D}(\mu_\star, \nu_\star)} \text{KL}(\mathbb{P}\|\mathbb{Q}), \tag{2}$$

where the coupling $\mathbb{P}$ belongs to the path space $\mathcal{D}(\mu_\star, \nu_\star) \subset C(\Omega, [0, T])$ with marginal measures $\mu_\star$ at time $t = 0$ and

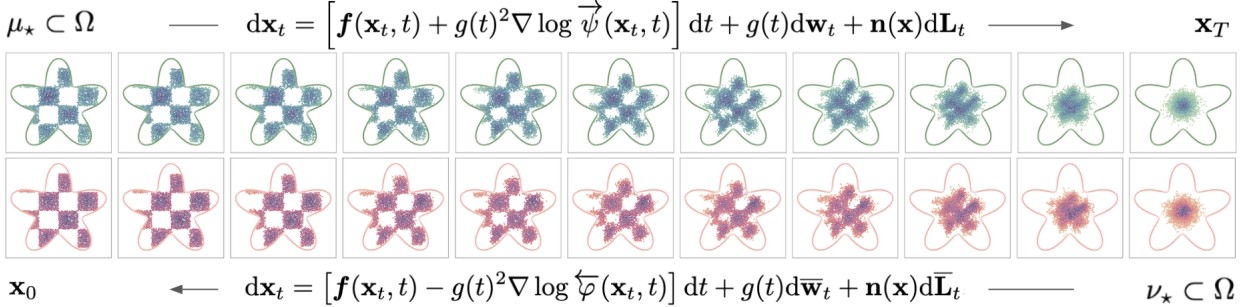

$$\mu_\star \subset \Omega \quad\text{---}\quad \mathrm{d}\mathbf{x}_t = \left[\boldsymbol{f}(\mathbf{x}_t, t) + g(t)^2 \nabla \log \overrightarrow{\psi}(\mathbf{x}_t, t)\right]\mathrm{d}t + g(t)\mathrm{d}\mathbf{w}_t + \mathbf{n}(\mathbf{x})\mathrm{d}\mathbf{L}_t \quad\longrightarrow\quad \mathbf{x}_T$$

$$\mathbf{x}_0 \quad\longleftarrow\quad \mathrm{d}\mathbf{x}_t = \left[\boldsymbol{f}(\mathbf{x}_t, t) - g(t)^2 \nabla \log \overleftarrow{\varphi}(\mathbf{x}_t, t)\right]\mathrm{d}t + g(t)\mathrm{d}\overline{\mathbf{w}}_t + \mathbf{n}(\mathbf{x})\mathrm{d}\overline{\mathbf{L}}_t \quad\text{---}\quad \nu_\star \subset \Omega$$

Figure 1: Constrained generative modeling via reflected forward-backward SDEs.

$\nu_\star$ at $t = T$; $\mathbb{Q}$ is the prior path measure, such as the measure induced by the path of the reflected Brownian motion or Ornstein-Uhlenbeck (OU) process. From the perspective of stochastic control, the dynamical SBP aims to minimize the cost along the reflected process

$$\inf_{\boldsymbol{u} \in \mathcal{U}} \mathbb{E}\left\{ \int_0^T \frac{1}{2}\|\boldsymbol{u}(\mathbf{x}_t, t)\|_2^2 \mathrm{d}t \right\}$$

$$\text{s.t. } \mathrm{d}\mathbf{x}_t = [\boldsymbol{f}(\mathbf{x}_t, t) + g(t)\boldsymbol{u}(\mathbf{x}_t, t)]\mathrm{d}t \qquad (3)$$
$$+ \sqrt{2\varepsilon}g(t)\mathrm{d}\mathbf{w}_t + \mathbf{n}(\mathbf{x}_t)\mathrm{d}\mathbf{L}_t,$$
$$\mathbf{x}_0 \sim \mu_\star, \ \mathbf{x}_T \sim \nu_\star, \ \mathbf{x}_t \in \Omega, \quad \text{for any } t \in [0, T]$$

where $\mathcal{U}$ is a set of control functions; $\varepsilon$ is the entropic regularizer for EOT; $\mathbf{n}(\mathbf{x})$ is an inner unit normal vector at $\mathbf{x} \in \partial\Omega$ and $\mathbf{0}$ for $\mathbf{x} \in \Omega$; the expectation follows from the density $\rho(\mathbf{x}, t)$. Simulation demos of the reflected SDEs are shown in Figure 2.

To derive the reflected FB-SDEs and training scheme, we first present standard assumptions on the regularity properties [Øksendal, 2003], as well as the smoothness of measure [Chen et al., 2022a,b] and boundary [Lamperski, 2021]:

**Assumption A1** (Regularity on drift and diffusion). *The drift $\boldsymbol{f}$ and diffusion term $g > 0$ satisfy the Lipschitz and linear growth condition.*

**Assumption A2** (Smooth boundary). *The domain $\Omega$ is bounded and has a smooth boundary.*

Extensions to general convex domains (with corners) are also studied in Lamperski [2021].

**Assumption A3** (Smooth measure). *The probability measures $\mu_\star$ and $\nu_\star$ are smooth such that the energy functions $U_\star = -\nabla \log \frac{\mathrm{d}\mu_\star}{\mathrm{d}\mathbf{x}}$ and $V_\star = -\nabla \log \frac{\mathrm{d}\nu_\star}{\mathrm{d}\mathbf{x}}$ are differentiable.*

## 4.1 REFLECTED FORWARD-BACKWARD STOCHASTIC DIFFERENTIAL EQUATIONS

Following the tradition in mechanics [Pavliotis, 2014], we rewrite the reflected SBP as follows

$$\inf_{\boldsymbol{u} \in \mathcal{U}} \int_0^T \int_\Omega \frac{1}{2}\rho\|\boldsymbol{u}\|_2^2 \mathrm{d}\mathbf{x}\mathrm{d}t$$

$$\text{s.t. } \frac{\partial\rho}{\partial t} + \nabla \cdot \mathbf{J}|_{\mathbf{x} \in \Omega} = 0, \ \langle \mathbf{J}, \mathbf{n} \rangle|_{\mathbf{x} \in \partial\Omega} = 0, \quad (4)$$

where $\mathbf{J}$ is the probability flux of continuity equation $\mathbf{J} \equiv \rho(\boldsymbol{f} + g\boldsymbol{u}) - \varepsilon g^2 \nabla\rho$ [Pavliotis, 2014].

We next solve the objectives with a Lagrangian multiplier: $\phi(\mathbf{x}, t)$. Applying the Stokes theorem with details presented in appendix A.1, we have

$$\mathcal{L}(\rho, \boldsymbol{u}, \phi) = \underbrace{\int_0^T \int_\Omega \left( \frac{1}{2}\rho\|\boldsymbol{u}\|_2^2 - \rho\frac{\partial\phi}{\partial t} - \langle \nabla\phi, \mathbf{J} \rangle \right)\mathrm{d}\mathbf{x}\mathrm{d}t}_{\overline{\mathcal{L}}(\rho, \boldsymbol{u}, \phi)}$$

$$(5)$$

$$+ \underbrace{\int_\Omega \phi\rho|_{t=0}^T \mathrm{d}\mathbf{x}}_{\text{constant term w.r.t. } \boldsymbol{u}} + \underbrace{\int_0^T \int_{\partial\Omega} \langle \mathbf{J}, \mathbf{n} \rangle \mathrm{d}\sigma(\mathbf{x})\mathrm{d}t}_{:=0 \text{ by Eq.(4)}}.$$

Minimizing $\mathcal{L}$ with respect to $\boldsymbol{u}$, we can obtain $\boldsymbol{u}^\star = g\nabla\phi$. Further applying the Cole-Hopf transform $\overrightarrow{\psi}(\mathbf{x}, t) = \exp\left(\frac{\phi(\mathbf{x},t)}{2\varepsilon}\right)$ and setting $\overline{\mathcal{L}}(\rho, \boldsymbol{u}^\star, \phi) = 0$, we derive the *backward Kolmogorov equation* with *Neumann boundary* conditions

$$\begin{cases} \frac{\partial\overrightarrow{\psi}}{\partial t} + \varepsilon g^2 \Delta\overrightarrow{\psi} + \langle \nabla\overrightarrow{\psi}, \boldsymbol{f} \rangle = 0 & \text{in } \Omega \\ \langle \nabla\overrightarrow{\psi}, \mathbf{n} \rangle = 0 & \text{on } \partial\Omega. \end{cases}$$

Next we define $\overleftarrow{\varphi} = \rho^\star / \overrightarrow{\psi}$, where $\rho^\star$ is the optimal density of Eq.(3) given $\boldsymbol{u}^\star$. We arrive at the *forward Kolmogorov equation* with the *Robin boundary* condition

$$\begin{cases} \partial_t\overleftarrow{\varphi} + \nabla \cdot (\overleftarrow{\varphi}\boldsymbol{f} - \varepsilon g^2 \nabla\overleftarrow{\varphi}) = 0 & \text{in } \Omega \\ \langle \overleftarrow{\varphi}\boldsymbol{f} - \varepsilon g^2 \nabla\overleftarrow{\varphi}, \mathbf{n} \rangle = 0 & \text{on } \partial\Omega. \end{cases}$$

Despite the elegance, solving PDEs in high dimensions often poses significant challenges due to the curse of dimensionality [Han et al., 2019]. To overcome these challenges, we resort to presenting a set of reflected FB-SDEs:

**Theorem 1.** *Consider a* Schrödinger (PDE) system *with Neumann and Robin boundary conditions*

$$\begin{cases} \frac{\partial \overrightarrow{\psi}}{\partial t} + \langle \nabla \overrightarrow{\psi}, \boldsymbol{f} \rangle + \varepsilon g^2 \Delta \overrightarrow{\psi} = 0 \\ \frac{\partial \overleftarrow{\varphi}}{\partial t} + \nabla \cdot (\overleftarrow{\varphi} \boldsymbol{f}) - \varepsilon g^2 \Delta \overleftarrow{\varphi} = 0 \end{cases} \quad (6)$$

*where* $\langle \nabla \overrightarrow{\psi}, \mathbf{n} \rangle|_{\mathbf{x} \in \partial \Omega} = 0, \langle \boldsymbol{f} \overleftarrow{\varphi} - \varepsilon g^2 \nabla \overleftarrow{\varphi}, \mathbf{n} \rangle|_{\mathbf{x} \in \partial \Omega} = 0.$

*Solving the PDE system gives rise to the reflected FB-SDEs*

$$d\mathbf{x}_t = \left[ \boldsymbol{f}(\mathbf{x}_t, t) + 2\varepsilon g(t)^2 \nabla \log \overrightarrow{\psi}(\mathbf{x}_t, t) \right] dt \quad (7a)$$
$$+ \sqrt{2\varepsilon} g(t) d\mathbf{w}_t + \mathbf{n}(\mathbf{x}) d\mathbf{L}_t, \ \mathbf{x}_0 \sim \mu_\star, \mathbf{x}_t \in \Omega,$$

$$d\mathbf{x}_t = \left[ \boldsymbol{f}(\mathbf{x}_t, t) - 2\varepsilon g(t)^2 \nabla \log \overleftarrow{\varphi}(\mathbf{x}_t, t) \right] dt \quad (7b)$$
$$+ \sqrt{2\varepsilon} g(t) d\overline{\mathbf{w}}_t + \mathbf{n}(\mathbf{x}) d\overline{\mathbf{L}}_t, \ \mathbf{x}_T \sim \nu_\star, \mathbf{x}_t \in \Omega.$$

*The connection to the probability flow ODE is also studied and presented in section A.2.*

### 4.2 LIKELIHOOD TRAINING

It is worth mentioning that the reflected FB-SDE (7) is not directly accessible due to the unknown control variables $(\nabla \log \overrightarrow{\psi}, \nabla \log \overleftarrow{\varphi})$. To tackle this issue, a standard tool is the (nonlinear) Feynman-Kac formula [Ma and Yong, 2007, Karatzas and Shreve, 1998], which leads to a stochastic representation.

**Proposition 1** (Feynman-Kac representation). *Assume assumptions A1-A2 hold.* $\overleftarrow{\varphi}$ *satisfies a PDE* (6) *and* $\mathbf{x}_t$ *follows from a diffusion* (7a). *Define* $\overrightarrow{y}_t \equiv \overrightarrow{y}(\mathbf{x}_t, t) = \log \overrightarrow{\psi}(\mathbf{x}_t, t)$ *and* $\overleftarrow{y}_t \equiv \overleftarrow{y}(\mathbf{x}_t, t) = \log \overleftarrow{\varphi}(\mathbf{x}_t, t).$

*Then* $\overleftarrow{y}_s$ *admits a stochastic representation*

$$\overleftarrow{y}_s = \mathbb{E}\left[ \overleftarrow{y}_T - \int_s^T \left( \zeta(\mathbf{x}_t, t) dt - d\overleftarrow{\mathbf{L}}_t \right) \Big| \mathbf{x}_s = \boldsymbol{x}_s \right],$$

*on* $\Omega \times [0, T]$; $\overrightarrow{\mathbf{z}}_t \equiv \overrightarrow{\mathbf{z}}(\mathbf{x}_t, t) = g\nabla \overrightarrow{y}_t$, $\overleftarrow{\mathbf{z}}_t \equiv \overleftarrow{\mathbf{z}}(\mathbf{x}_t, t) = g\nabla \overleftarrow{y}_t$, $d\overleftarrow{\mathbf{L}}_t = \frac{1}{g}\langle \overleftarrow{\mathbf{z}}_t, \mathbf{n}_t \rangle d\mathbf{L}_t$, $\zeta(\mathbf{x}_t, t) = \frac{1}{2}\|\overleftarrow{\mathbf{z}}_t\|_2^2 + \nabla \cdot \left( \frac{1}{g}\overleftarrow{\mathbf{z}}_t - \boldsymbol{f} \right) + \langle \overleftarrow{\mathbf{z}}_t, \overrightarrow{\mathbf{z}}_t \rangle.$

**Sketch of proof** The proof primarily relies on Theorem 3 from Chen et al. [2022b] and applies (generalized) Itô's lemma to $\overleftarrow{y}_t$ using (6) and (7a). The difference is to incorporate the generalized Itô's lemma [Bubeck et al., 2018, Lamperski, 2021] to address the local time of $\mathbf{x}_t$ at the boundary $\partial \Omega$. Subsequently, our analysis establishes that $\overleftarrow{y}_s - \int_{s_1}^s \zeta(\mathbf{x}_t, t) dt - d\overleftarrow{\mathbf{L}}_t$, where $s \in [s_1, T]$, is a martingale in the domain $\Omega$. $\square$

A direct application of the proposition is to obtain the log-likelihood $\overleftarrow{y}_0$ given data points $\mathbf{x}_0$. With parametrized models $(\overrightarrow{\mathbf{z}}_t^\theta, \overleftarrow{\mathbf{z}}_t^\omega)$ to approximate $(\overrightarrow{\mathbf{z}}_t, \overleftarrow{\mathbf{z}}_t)$, we can optimize the backward score function $\overleftarrow{\mathbf{z}}_t^\omega$ through the forward loss function $\mathcal{L}(\mathbf{x}_0; \omega)$ in Algorithm 1. Regarding the forward-score estimation, similar to Theorem 11 [Chen et al., 2022b], the symmetric property of the reflected SB also enables to optimize $\overrightarrow{\mathbf{z}}_t$ via the backward loss function $\mathcal{L}(\mathbf{x}_T; \theta)$.

By the data processing inequality, our loss function provides a lower bound of the log-likelihood, which resembles the evidence lower bound (ELBO) in variational inference [Song et al., 2021a]. We can expect a smaller variational gap given more accurate parametrized models.

When the domain is taken to be $\Omega = \mathbb{R}^d$, the aforementioned solvers become equivalent to the loss function (18-19) presented in Chen et al. [2022b].

### 4.3 CONNECTIONS TO THE IPF ALGORITHM

Similar in spirit to Theorem 3 of Song et al. [2021a], Algorithm 1 results in an elegant half-bridge solver ($\mu_\star \rightarrow \nu_\star$ v.s. $\mu_\star \leftarrow \nu_\star$) to approximate the primal formulation [Nutz, 2022] of the dynamic Schrödinger bridge (2) [De Bortoli et al., 2021, Vargas et al., 2021]:

**Dynamic Primal IPF** $\mathbb{P}_{2k} = \underset{\mathbb{P} \in \mathcal{D}(\cdot, \nu_\star)}{\arg\min} \text{KL}(\mathbb{P} \| \mathbb{P}_{2k-1}),$
$$\mathbb{P}_{2k+1} = \underset{\mathbb{P} \in \mathcal{D}(\mu_\star, \cdot)}{\arg\min} \text{KL}(\mathbb{P} \| \mathbb{P}_{2k}), \quad (8)$$

which is also known as the dynamic IPF algorithm (also known as Sinkhorn algorithm) [Ruschendorf, 1995, De Bortoli et al., 2021]. Consider the disintegration of the path measure $\mathbb{P} = \pi \otimes \mathbb{P}^{\mu_\star, \nu_\star}$

$$\mathbb{P}(\cdot) = \iint_{\Omega^2} \mathbb{P}^{\mathbf{x}_0, \mathbf{x}_T}(\cdot) \pi(d\mathbf{x}_0, d\mathbf{x}_T), \quad (9)$$

where $\mathbb{P}^{\mathbf{x}_0, \mathbf{x}_T} \in \mathbb{P}^{\mu_\star, \nu_\star}$ is a diffusion bridge from $\mathbf{x}_0 = \mathbf{x}_0$ to $\mathbf{x}_T = \mathbf{x}_T$, $\pi \in \Pi(\mu_\star, \nu_\star)$ and the product space $\Pi(\mu_\star, \nu_\star) \subset \Omega^2$ denotes the space of couplings with the first and second marginals following from $\mu_\star$ and $\nu_\star$, respectively. Now project the path space $\mathcal{D}$ to the product space $\Pi$. We have the static IPF algorithm in the primal formulation:

**Static Primal IPF** $\pi_{2k} = \underset{\pi \in \Pi(\cdot, \nu_\star)}{\arg\min} \text{KL}(\pi \| \pi_{2k-1}),$
$$\pi_{2k+1} = \underset{\pi \in \Pi(\mu_\star, \cdot)}{\arg\min} \text{KL}(\pi \| \pi_{2k}). \quad (10)$$

## 5 CONVERGENCE ANALYSIS VIA ENTROPIC OPTIMAL TRANSPORT

The dynamic IPF algorithm offers an efficient training scheme to fit marginals in high-dimensional problems. However, the understanding of the convergence remains unclear

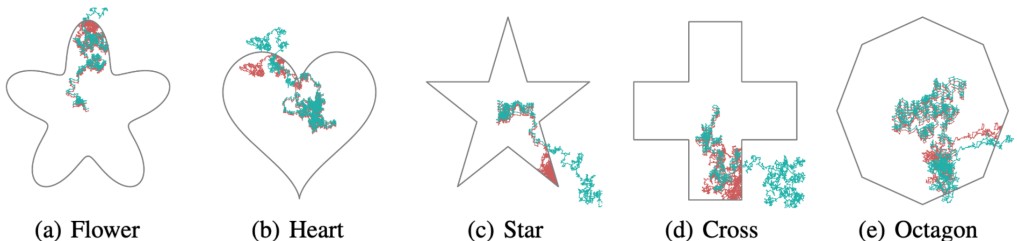

| (a) Flower | (b) Heart | (c) Star | (d) Cross | (e) Octagon |

Figure 2: Reflected OU processes (**reflected** v.s. unconstrained), driven by the same Brownian motion, excluding the reflections. All boundary curves have properly defined unit vectors.

---

**Algorithm 1** One iteration of the backward-forward score function solver to optimize $(\overrightarrow{\mathbf{z}}_t^\theta, \overleftarrow{\mathbf{z}}_t^\omega)$ with the reflection implemented in Algorithm 4. We cache the trajectories following De Bortoli et al. [2021] to avoid expensive computational graphs. In practice, $\mathbb{E}[\log \overleftarrow{y}_T]$ and $\mathbb{E}[\log \overrightarrow{y}_0]$ are often omitted to facilitate training [Chen et al., 2022b].

---

$$\mathcal{L}(\mathbf{x}_0; \omega) = -\int_0^T \mathbb{E}_{\mathbf{x}_t \sim (7a)} \left[ \left( \frac{1}{2} \|\overleftarrow{\mathbf{z}}_t^\omega\|_2^2 + g\nabla \cdot \overleftarrow{\mathbf{z}}_t^\omega + \langle \overrightarrow{\mathbf{z}}_t^\theta, \overleftarrow{\mathbf{z}}_t^\omega \rangle \right) \mathrm{d}t + \mathrm{d}\overleftarrow{\mathbf{L}}_t^\omega \Big| \mathbf{x}_0 = \mathbf{x}_0 \right]$$

$$\mathcal{L}(\mathbf{x}_T; \theta) = -\int_0^T \mathbb{E}_{\mathbf{x}_t \sim (7b)} \left[ \left( \frac{1}{2} \|\overrightarrow{\mathbf{z}}_t^\theta\|_2^2 + g\nabla \cdot \overrightarrow{\mathbf{z}}_t^\theta + \langle \overleftarrow{\mathbf{z}}_t^\omega, \overrightarrow{\mathbf{z}}_t^\theta \rangle \right) \mathrm{d}t + \mathrm{d}\overrightarrow{\mathbf{L}}_t^\theta \Big| \mathbf{x}_T = \mathbf{x}_T \right],$$

where $\mathrm{d}\overleftarrow{\mathbf{L}}_t^\omega = \frac{1}{g}\langle \overleftarrow{\mathbf{z}}_t^\omega, \mathbf{n}_t \rangle \mathrm{d}\mathbf{L}_t$ and $\mathrm{d}\overrightarrow{\mathbf{L}}_t^\theta = \frac{1}{g}\langle \overrightarrow{\mathbf{z}}_t^\theta, \mathbf{n}_t \rangle \mathrm{d}\overline{\mathbf{L}}_t$. (7a) (respectively, (7b)) is approximated via $\overrightarrow{\mathbf{z}}_t^\theta$ (respectively, $\overleftarrow{\mathbf{z}}_t^\omega$).

---

to the machine learning community. To get around this issue, we leverage the progress from the static optimal transport on bounded domains and costs [Carlier, 2022, Chen et al., 2016, Deligiannidis et al., 2021].

Our analysis is illustrated as follows: We first draw connections between dynamic and static (primal) IPFs by projecting the path space $\mathcal{D}$ to the product space $\Pi$ and then show the equivalence between the dual and primal formulations. Next, we perturb the marginals (in terms of energy functions) and show the approximate linear convergence of the dual, potential, and then static couplings. The convergence of dynamic couplings can be expected given a reasonable estimate of diffusion bridge.

Dynamic Primal IPF (8) $\xleftrightarrow[\text{Projection}]{\text{Disintegration}}$ Static Primal IPF (10)

$\xleftrightarrow[\text{Lemma 1}]{\text{Equivalence(B.5)}}$ Static Dual IPF (14)

### 5.1 EQUIVALENCE BETWEEN DYNAMIC SBP AND STATIC SBP

Assuming the solutions exist, the disintegration of measures implies that the equivalence of solutions between the dynamic and static SBPs [Léonard, 2014]:

**Dynamic SBP** $\quad \mathbb{P}_\star = \underset{\mathbb{P} \in \mathcal{D}(\mu_\star, \nu_\star)}{\arg\min} \mathrm{KL}(\mathbb{P}\|\mathbb{Q})$

$\iff \pi_\star = \underset{\pi \in \Pi(\mu_\star, \nu_\star)}{\arg\min} \mathrm{KL}(\pi\|\mathcal{G}),$ **Static**

where $\pi$ (respectively, $\mathcal{G}$) is the projection of the path measure $\mathbb{P}$ (respectively, $\mathbb{Q}$) on the product space at $t = 0$ and

$T$; $\mathrm{d}\mathcal{G} \propto e^{-c_\varepsilon}\mathrm{d}(\mu_\star \otimes \nu_\star)$; $c_\varepsilon$ is a cost function. Both the dynamic and static SBP formulations yield structure properties (see the Born's formula in Léonard [2014]) and enables to represent Schrödinger bridges $\mathbb{P}_\star$ and $\pi_\star$ using Schrödinger potentials $\varphi_\star$ and $\psi_\star$:

**Dynamic Struture** $\quad \mathrm{d}\mathbb{P}_\star = e^{\varphi_\star(\mathbf{x})+\psi_\star(\mathbf{y})}\mathrm{d}\mathbb{Q}$ (11)

$\iff \mathrm{d}\pi_\star(\mathbf{x}, \mathbf{y}) = e^{\varphi_\star(\mathbf{x})+\psi_\star(\mathbf{y})}\mathrm{d}\mathcal{G}.$ **Static**

Moreover, the summation $\varphi_\star \oplus \psi_\star$ is unique such that $(\varphi_\star + a) \oplus (\psi_\star - a)$ is also viable for any $a$.

This static structural representation establishes a connection between the static SBP and entropic optimal transport (EOT) with a unit entropy regularizer [Chen et al., 2023], and the latter results in an efficient scheme to compute the optimal coupling:

$$\inf_{\pi \in \Pi(\mu_\star, \nu_\star)} \iint_{\Omega^2} c_\varepsilon(\mathbf{x}, \mathbf{y})\pi(\mathrm{d}\mathbf{x}, \mathrm{d}\mathbf{y}) + \mathrm{KL}(\pi\|\mu_\star \otimes \nu_\star).$$

### 5.2 DUALITY FOR SCHRÖDINGER BRIDGES AND APPROXIMATIONS

The Schrödinger bridge is a constrained optimization problem and possesses a computation-friendly dual formulation. Moreover, the duality gap is zero under probability measures [Léonard, 2001].

**Lemma 1** (Duality [Nutz, 2022])**.** *Given assumptions A1-A3, the dual via potentials $(\varphi, \psi)$ follows*

$$\min_{\pi \in \Pi(\mu_\star, \nu_\star)} \mathrm{KL}(\pi|\mathcal{G}) = \max_{\varphi, \psi} G(\varphi, \psi),$$ (12)

*where* $G(\varphi, \psi) := \mu_\star(\varphi) + \nu_\star(\psi) - \iint_{\Omega^2} e^{\varphi \oplus \psi} \mathrm{d}\mathcal{G} + 1$, $\mu_\star(\varphi) = \int_\Omega \varphi \mathrm{d}\mu_\star$, $\nu_\star(\psi) = \int_\Omega \psi \mathrm{d}\nu_\star$, $\varphi \in L^1(\mu_\star)$, *and* $\psi \in L^1(\nu_\star)$.

An effective solver is to maximize the dual $G$ via $\varphi_{k+1} = \arg\max_{\varphi \in L^1(\mu_\star)} G(\varphi, \psi_k)$ and $\psi_{k+1} = \arg\max_{\psi \in L^1(\nu_\star)} G(\varphi_{k+1}, \psi)$ alternatingly. From a geometric perspective, alternating maximization corresponds to alternating projections (detailed in Appendix B.4)

$$\varphi_{k+1} = \arg\max_{\varphi \in L^1(\mu_\star)} G(\varphi, \psi_k) \tag{13a}$$

$$\implies \text{ the first marginal of } \pi(\varphi_{k+1}, \psi_k) \text{ is } \mu_\star,$$

$$\psi_{k+1} = \arg\max_{\psi \in L^1(\nu_\star)} G(\varphi_{k+1}, \psi) \tag{13b}$$

$$\implies \text{ the second marginal of } \pi(\varphi_{k+1}, \psi_{k+1}) \text{ is } \nu_\star.$$

The marginal properties of the coupling implies the Schrödinger equation [Nutz and Wiesel, 2022]

$$\varphi_\star(\mathbf{x}) = -\log \int_\Omega e^{\psi_\star(\mathbf{y}) - c_\varepsilon(\mathbf{x}, \mathbf{y})} \nu_\star(\mathrm{d}\mathbf{y}),$$

$$\psi_\star(\mathbf{y}) = -\log \int_\Omega e^{\varphi_\star(\mathbf{x}) - c_\varepsilon(\mathbf{x}, \mathbf{y})} \mu_\star(\mathrm{d}\mathbf{x}).$$

Since the Schrödinger potential functions $(\psi_\star, \varphi_\star)$ are not known *a priori*, the dual formulation of the static IPF algorithm was proposed to solve the alternating projections as follows:

**Static Dual IPF** : $\psi_k(\mathbf{y}) = -\log \int_\Omega e^{\varphi_k(\mathbf{x}) - c_\varepsilon(\mathbf{x}, \mathbf{y})} \mu_\star(\mathrm{d}\mathbf{x}),$

$$\varphi_{k+1}(\mathbf{x}) = -\log \int_\Omega e^{\psi_k(\mathbf{y}) - c_\varepsilon(\mathbf{x}, \mathbf{y})} \nu_\star(\mathrm{d}\mathbf{y}). \tag{14}$$

The equivalence between the primal IPF and dual IPF is further illustrated in Appendix B.5.

However, given a limited computational budget, projecting to the ideal measure $\mu_\star$ (or $\nu_\star$) in Eq.(13) at each iteration may not be practical. Instead, some close approximation $\mu_{\star,k+1}$ (or $\nu_{\star,k}$) is used at iteration $2k+1$ (or $2k$) via Gaussian processes [Vargas et al., 2021] or neural networks [De Bortoli et al., 2021, Chen et al., 2022b]. Therefore, one may resort to an approximate marginal that still achieves reasonable accuracy:

$$\mu_{2k+1} = \mu_{\star,k+1} \approx \mu_\star, \quad \nu_{2k} = \nu_{\star,k} \approx \nu_\star. \tag{15}$$

We refer to the IPF algorithm with approximate marginals as approximate IPF (aIPF) and present the static dual formulation of aIPF in Algorithm 2. The difference between IPF and aIPF is detailed in Figure 3. The structure representation (11) can be naturally extended based on approximate

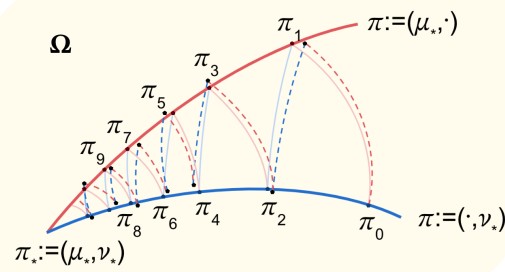

Figure 3: IPF v.s. aIPF. The approximate (or exact) projections are highlighted through the dotted (or solid) lines.

marginals and is also studied by Deligiannidis et al. [2021]

$$\mathrm{d}\pi_{2k} = e^{\varphi_k \oplus \psi_k - c_\varepsilon} \mathrm{d}(\mu_{\star,k} \otimes \nu_{\star,k}),$$

$$\mathrm{d}\pi_{2k-1} = e^{\varphi_k \oplus \psi_{k-1} - c_\varepsilon} \mathrm{d}(\mu_{\star,k} \otimes \nu_{\star,k-1}), \tag{16}$$

where $\pi_k$ is the approximate coupling at iteration $k$. By the structural properties in Eq.(11), the representation also applies to the dynamic settings, which involves the computation of the static IPF, followed by its integration with a diffusion bridge [Eckstein and Nutz, 2022].

---

**Algorithm 2** One iteration of aIPF (static). The static coupling $\pi_k$ can be recovered by the structural representation in (16); the dynamic coupling $\mathbb{P}_k = \iint_{\Omega^2} \mathbb{P}_k^{\mathbf{x}_0, \mathbf{x}_T}(\cdot) \pi_k(\mathbf{x}_0, \mathbf{x}_T)$ can be solved by further learning a diffusion bridge $\mathbb{P}_k^{\mathbf{x}_0, \mathbf{x}_T}$.

$$\psi_k(\mathbf{y}) = -\log \int_\Omega e^{\varphi_k(\mathbf{x}) - c_\varepsilon(\mathbf{x}, \mathbf{y})} \mu_{\star,k}(\mathrm{d}\mathbf{x}),$$

$$\varphi_{k+1}(\mathbf{x}) = -\log \int_\Omega e^{\psi_k(\mathbf{y}) - c_\varepsilon(\mathbf{x}, \mathbf{y})} \nu_{\star,k}(\mathrm{d}\mathbf{y}). \tag{17}$$

---

### 5.3 CONVERGENCE OF COUPLINGS WITH BOUNDED DOMAIN

Despite the rich literature on the analysis of SBP on bounded domains [Chen et al., 2016], most of them are not applicable to practical scenarios where exact marginals are not available. To fill this gap, we extend the linear convergence with perturbed marginals. The key to our proof is the strong convexity of the dual (12). To quantify the convergence, similar to De Bortoli [2022], we introduce an assumption to control the perturbation of the marginals such that:

**Assumption A4** (Marginal perturbation). $U_k = \nabla \log \frac{\mathrm{d}\mu_{\star,k}}{\mathrm{d}\mathbf{x}}$ *and* $V_k = \nabla \log \frac{\mathrm{d}\nu_{\star,k}}{\mathrm{d}\mathbf{x}}$ *are the approximate energy functions at the $k$-th iteration and are $\epsilon$-close to energy functions $U_\star$ and $V_\star$*

$$\|U_k(\mathbf{x}) - U_\star(\mathbf{x})\|_2 \le \epsilon(1 + \|\mathbf{x}\|_2),$$

$$\|V_k(\mathbf{x}) - V_\star(\mathbf{x})\|_2 \le \epsilon(1 + \|\mathbf{x}\|_2), \quad \forall \mathbf{x} \in \Omega.$$

Note that the Lipschitz cost function on $\Omega^2$ is also a standard assumption [Deligiannidis et al., 2021]. It is not required here by Assumption A1, which leads to a smooth transition kernel and cost function.

Recall the connections between dynamic primal IPF and static dual IPF, we know $\epsilon$ mainly depends on the score-function $(\overrightarrow{\mathbf{z}}_t^\theta, \overleftarrow{\mathbf{z}}_t^\omega)$ estimations [Song et al., 2021a] and numerical discretizations. More concrete connections between them will be left as future work. In addition, the errors in the two marginals don't have to be the same, and we use a unified $\epsilon$ mainly for analytical convenience.

Moreover, we use the same domain $\Omega$ for both marginals to be consistent with our algorithm in Section 4. The proof can be extended to different domains X and Y for $\mu_\star$ and $\nu_\star$.

**Approximately linear convergence and proof sketches**
We first follow Carlier [2022], Nutz [2022], Marino and Gerolin [2020] to build a *centered* aIPF algorithm in Algorithm 3 with scaled potential functions $\bar\varphi_k$ and $\bar\psi_k$ such that $\mu_\star(\bar\varphi_k) = 0$. Since the summations of the potentials $\varphi_\star$ and $\psi_\star$ are unique by (11), the *centering* operation doesn't change the dual objective but ensures that the aIPF iterates are uniformly bounded in Lemma 4 by the decomposition

$$\|\bar\varphi \oplus \bar\psi\|^2_{L^2(\mu_\star \otimes \nu_\star)} = \|\bar\varphi\|^2_{L^2(\mu_\star)} + \|\bar\psi\|^2_{L^2(\nu_\star)} \text{ if } \mu_\star(\bar\varphi) = 0.$$

How to ensure centering with perturbed marginals in Algorithm 3 is crucial and one major novelty in our proof. We next exploit the *strong convexity* of the exponential function $e^x$ w.r.t. the concave dual. We obtain an auxiliary result regarding the convergence of the dual and the potentials.

**Lemma 2** (Convergence of the Dual and Potentials). *Let $(\bar\varphi_k, \bar\psi_k)_{k\geq 0}$ be the iterates of a variant of Algorithm 2. Given assumptions A1-A4 with small enough marginal perturbations $\epsilon$, we have*

$$G(\bar\varphi_\star, \bar\psi_\star) - G(\bar\varphi_k, \bar\psi_k) \lesssim (1 - e^{-24\|c_\varepsilon\|_\infty})^k + e^{24\|c_\varepsilon\|_\infty}\epsilon,$$
$$\|\bar\varphi_\star - \bar\varphi_k\|_{L^2(\mu_\star)} + \|\bar\psi_\star - \bar\psi_k\|_{L^2(\nu_\star)}$$
$$\lesssim e^{3\|c_\varepsilon\|_\infty}(1 - e^{-24\|c_\varepsilon\|_\infty})^{k/2} + e^{15\|c_\varepsilon\|_\infty}\epsilon^{1/2}.$$

Since the centering operation doesn't change the structure property (11), we are able to analyze the convergence of the static couplings. Motivated by Theorem 3 of Deligiannidis et al. [2021], we exploit the structural property (11) to estimate the $\mathbf{W}_1$ distance based on its dual formulation.

**Theorem 2** (Convergence of Static Couplings). *Given assumptions A1-A4 with small marginal perturbations $\epsilon$, the iterates of the couplings $(\pi_k)_{k\geq 0}$ in Algorithm 2 satisfy the following result*

$$\mathbf{W}_1(\pi_k, \pi_\star)$$
$$\leq O\left(e^{9\|c_\varepsilon\|_\infty}(1 - e^{-24\|c_\varepsilon\|_\infty})^{k/2} + e^{21\|c_\varepsilon\|_\infty}\epsilon^{1/2}\right).$$

Such a result provides the worst-case guarantee on the convergence of the static couplings $\pi_k$. For example, to obtain a $\epsilon_\star$-$\mathbf{W}_1$ distance, we can run $\Omega(e^{24\|c_\varepsilon\|_\infty}(\|c_\varepsilon\|_\infty - \log(\epsilon_\star \wedge 1)))$ iterations to achieve the goal. Recall that $c_\varepsilon = c/\varepsilon$ [Chen et al., 2023], a large entropic-regularizer $\varepsilon$ may be needed in practice to yield reasonable performance, which also leads to specific tuning guidance on $\varepsilon$.

Our proof employs a non-geometric method to show the uniform in time stability, w.r.t. the marginals. Unlike the elegant approach [Deligiannidis et al., 2021] based on the Hilbert-Birkhoff projective metric [Chen et al., 2016], ours does not require advanced tools and may be more friendly to readers.

Recall the bridge representation in Eq.(9), we have $\mathbf{W}_1(\pi_k \otimes \mathbb{P}_k^{\mu_\star, \nu_\star}, \pi_\star \otimes \mathbb{P}_\star^{\mu_\star, \nu_\star}) \leq \mathbf{W}_1(\pi_k, \pi_\star) + \mathbf{W}_1(\mathbb{P}_k^{\mu_\star, \nu_\star}, \mathbb{P}_\star^{\mu_\star, \nu_\star})$. Assume the same assumptions as in Theorem 2, we arrive at the final result:

**Proposition 2** (Convergence of Dynamic Couplings). *The iterates of the dynamic couplings $(\mathbb{P}_k)_{k\geq 0}$ in Algorithm 1 satisfy the following result*

$$\mathbf{W}_1(\mathbb{P}_k, \mathbb{P}_\star) \leq O(e^{9\|c_\varepsilon\|_\infty}(1 - e^{-24\|c_\varepsilon\|_\infty})^{k/2})$$
$$+ O(e^{21\|c_\varepsilon\|_\infty}\epsilon^{1/2}) + \mathbf{W}_1(\mathbb{P}_k^{\mu_\star, \nu_\star}, \mathbb{P}_\star^{\mu_\star, \nu_\star}).$$

The result paves the way for understanding the general convergence of the dynamic IPF algorithm by incorporating a proper approximation of the diffusion bridge [Heng et al., 2022].

# 6 EMPIRICAL SIMULATIONS

## 6.1 GENERATION OF 2D SYNTHETIC DATA

We first employ the reflected SB algorithm to generate three synthetic examples: checkerboard and Gaussian mixtures from a Gaussian prior and spiral from a moon prior. The domains are defined to be flower, octagon, and heart, where all boundary points are defined to have proper unit-vectors. We follow Chen et al. [2022b] and adopt a U-net to model $(\overrightarrow{\mathbf{z}}_t^\theta, \overleftarrow{\mathbf{z}}_t^\omega)$. We chose RVP-SDE as the base simulator from time 0 to $T = 1$, where the dynamics are discretized into 100 steps.

Our generated examples are presented in Figure 1 and 4. We see that all the data are generated smoothly from the prior and the forward and backward process matches with each other elegantly. To the best of our knowledge, this is the first algorithm (with OT guarantees) that works on custom domains. Other related work, such as Lou and Ermon [2023], mainly focuses on hypercubes in computer vision. We also visualize the forward-backward policies $\overleftarrow{\mathbf{z}}_t^\omega$ and $\overrightarrow{\mathbf{z}}_t^\theta$ in Figure 4. Our observations reveal that the forward vector fields $\overrightarrow{\mathbf{z}}_t^\theta$ demonstrate substantial nonlinearity when compared to the linear forward policy in SGMs, and furthermore, the

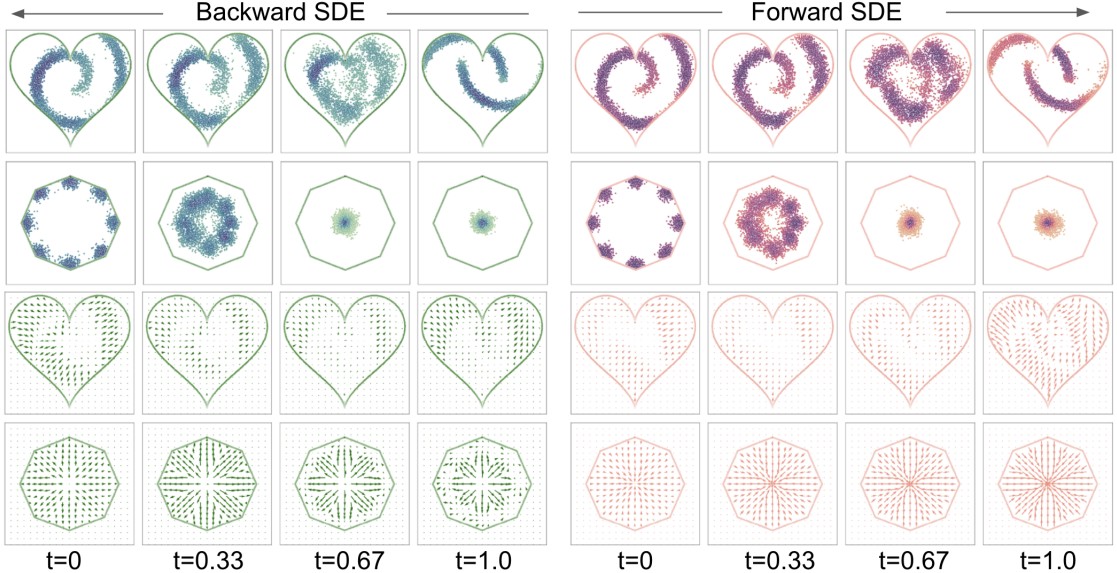

Figure 4: Demo of generative samples (top) and vector fields (bottom) based on Reflected SB.

forward vector fields exhibit pronounced dissimilarity when compared to the backward vector fields $\overleftarrow{\mathbf{z}}_t^\omega$.

## 6.2 GENERATION OF IMAGE DATA

We test our method on large-scale image datasets using CIFAR-10 and ImageNet $64\times64$. As the RGB value is between $[0, 1]$, we naturally select the domain as $\Omega = [0, 1]^d$, where $d = 3 \times 32 \times 32$ for the CIFAR-10 task and $d = 3 \times 64 \times 64$ for the ImageNet task. It is known that the SB system can be initialized with score-based generative models [Chen et al., 2022b] and the warm-up study for reflected SB is presented in Appendix C.2. We choose RVE-SDE as the prior path measure. The prior distribution of $\nu_\star$ is the uniform distribution on $\Omega$. The SDE is discretized into 1000 steps. In both scenarios, images are generated unconditionally, and the quality of the samples is evaluated using Frechet Inception Distance (FID) over 50,000 samples. The forward score function is modeled using U-net structure; the backward score function uses NCSN++ [Song et al., 2021b] for the CIFAR-10 task and ADM [Dhariwal and Nichol, 2022] for the ImageNet task. Details of the experiments are shown in Appendix C.

We have included baselines for both constrained and unconstrained generative models and summarized the experimental results in Table 1. While our model may not surpass the state-of-the-art models, the minor improvement over the unconstrained SB-FBSDE [Chen et al., 2022b] underscores the effectiveness of the reflection operation. Moreover, the experiments verify the scalability of the reflected model and the training process is consistent with the findings in Lou and Ermon [2023], where the reflection in cube domains is easy to implement and the generation becomes more stable. Sample outputs are showcased in Figure 5 (including

| CIFAR-10 | Constrained | OT | NLL | FID |
|---|---|---|---|---|
| MCSN++ [Song et al., 2021b] | No | No | 2.99 | 2.20 |
| DDPM [Ho et al., 2020] | No | No | 3.75 | 3.17 |
| SB-FBSDE [Chen et al., 2022b] | No | Yes | - | 3.01 |
| Reflected SGM [Lou and Ermon, 2023] | Yes | No | 2.68 | 2.72 |
| **Ours** | Yes | Yes | 3.08 | 2.98 |
| | | | | |
| ImageNet $64\times64$ | | | | |
| PGMGAN [Armandpour et al., 2021] | No | No | – | 21.73 |
| GLIDE [Li et al., 2023] | No | No | – | 29.18 |
| GRB [Park and Shin, 2022] | No | No | – | 26.57 |
| **Ours** | Yes | Yes | 3.20 | 23.95 |

Table 1: Evaluation of generative models on image data.

MNIST), with additional figures available in Appendix C. Notably, our generated samples exhibit diversity and are visually indistinguishable from real data.

## 6.3 GENERATION IN THE SIMPLEX DOMAIN

Alongside the irregular domains illustrated in Figure 2 and the hypercube for image generation, we implement the method on the high-dimensional *projected simplex*. A $d$-projected simplex is defined as $\bar{\Delta}_d := \{\boldsymbol{x} \in \mathbb{R}^d : \sum_i \boldsymbol{x}_i \leq 1, \boldsymbol{x}_i \geq 0\}$. Our method relies on reflected diffusion process instead of using diffeomorphic mapping (stick breaking) as in Lou and Ermon [2023]. As a comparison, we replicate the generative process using diffeomorphic mapping as well.

The data is created by collecting the image classification scores of Inception v3 from the last softmax layer with 1008 dimension. All the data fit into the projected simplex

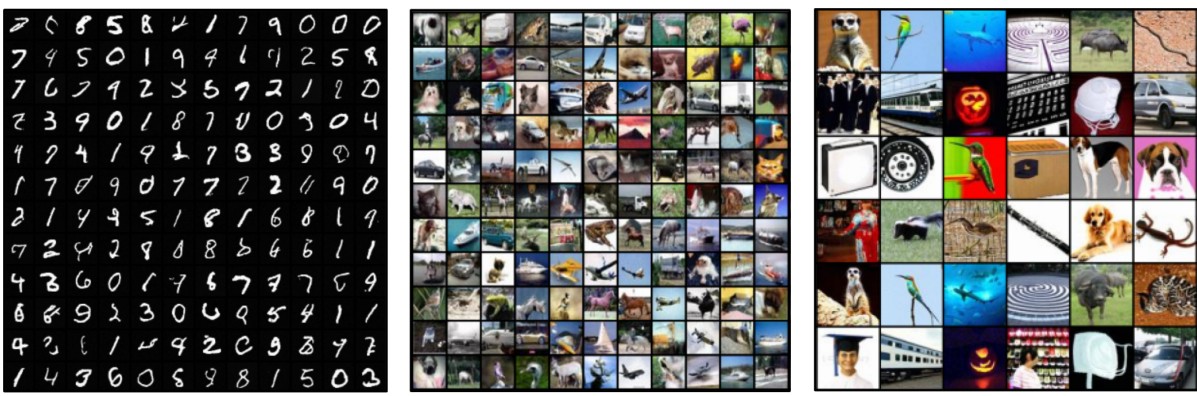

Figure 5: Samples via reflected SB on MNIST (left), CIFAR10 (middle), and ImageNet 64 (right).

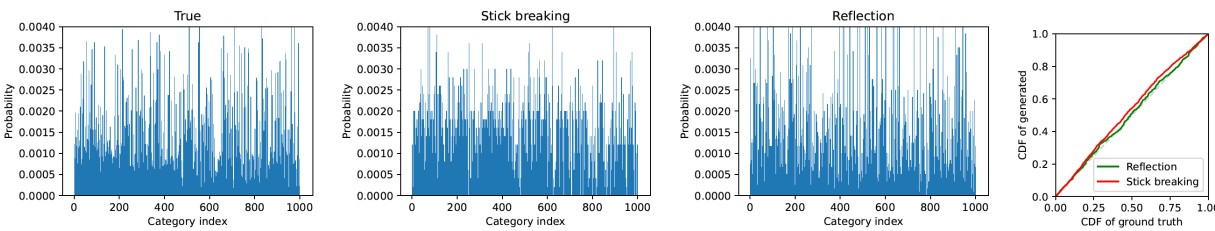

Figure 6: Generations of high-dimensional projected simplex. The results compare the reflection-based and stick-breaking based methods.

$\bar{\Delta}_{1008}$. The Inception model is loaded from a pretrained checkpoint[*], and the classification task is performed on the $64 \times 64$ Imagenet validation dataset of 50,000 images. The neural network of the score function is composed of 6 dense layers with 512 latent nodes. In every diffusion step, we use the reflection operator described in Algorithm 4 to constrain the data within the projected simplex. The alternative method is using stick breaking method to constrain the diffusion process. The transformation includes the mapping $[f(\boldsymbol{x})]_i = \boldsymbol{x}_i \prod_{j=i+1}^{d}(1 - \boldsymbol{x}_j)$ and the inverse mapping $[f^{-1}(\boldsymbol{y})]_i = \frac{\boldsymbol{y}_i}{1 - \sum_{j=i+1}^{d} \boldsymbol{y}_j}$. In every diffusion step, it first maps the data into an unit cube domain using reflection, then uses the forward transformation to map it within the projected simplex.

The results are shown in Figure 6. We compare the generated distribution of the most likely classes. The category index is in the same order of the pre-trained model's output. The last plot in Figure 6 compares the cumulative distribution of the ground truth and generated distribution, providing a cleaner view of the comparison. The curve closely follows the diagonal in the CDF comparison, signifying a strong alignment between the true data distribution and the distribution derived from the generative model. The result using diffeomorphic mapping is shown in Figure 6. By comparing the CDF comparison plots of two methods, the reflec-

tion based method outperforms the diffeomorphic based method, where the latter suffers from visible bias of the distribution due to the analytic blowups at edges/corners at edges/corners.

## 7 CONCLUSION

Reflected diffusion models, which are motivated by thresholding techniques, introduce explicit score-matching loss through reflected Brownian motion. Traditionally, these models are applied to hypercube-related domains and necessitate specific diffeomorphic mappings for extension to other domains. To enhance generality with optimal transport guarantees, we introduce the Reflected Schrödinger Bridge, which employs reflected forward-backward stochastic differential equations with Neumann and Robin boundary conditions. We establish connections between dynamic and static IPF algorithms in both primal and dual formulations. Additionally, we provide an approximate linear convergence analysis of the dual, potential, and couplings to deepen our understanding of the dynamic IPF algorithm. Empirically, our algorithm can be applied to any smooth domains using RVE-SDE and RVP-SDE. We evaluate its performance on 2D synthetic examples and standard image benchmarks, underscoring its competitiveness in constrained generative modeling. In future research, our focus includes integrating nonlinear diffusion based on importance sampling [Deng et al., 2022a,b] to further expedite the generation process.

---

[*]https://github.com/mseitzer/pytorch-fid/releases/download/fid_weights

# ACKNOWLEDGEMENTS

Q. Feng is partially supported by the National Science Foundation under grant DMS-2306769.

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
