# OpenReview forum: "Reflected Schr\"odinger Bridge for Constrained Generative Modeling"
_auai.org/UAI/2024/Conference — UAI 2024 oral_

### Official Review · Reviewer_StHn · 2024-02-29

**Q2-1 Originality-Novelty:** 4
**Q2-2 Correctness-Technical Quality:** 4
**Q2-5 Clarity Of Writing:** 4

**Q1 Summary And Contributions:**

The authors provide a new algorithm for solving Schrodinger bridges on bounded domains, which is particularly useful for solving SBPs under constraints, on the way the authors also prove several convergence remarks for their proposed methodology and do a thorough empirical evaluation demonstrating the strengths of their approach

**Q2-3 Extent To Which Claims Are Supported By Evidence:**

4: Excellent: all claims are supported by very convincing evidence (in the form of comprehensive experimental evaluation, rigorous mathematical proofs, detailed (pseudo-)code, precise references, well-motivated and realistic assumptions) and the authors deliver what they promise.

**Q2-4 Reproducibility:**

4: Excellent: key resources (e.g. proofs, code, data) are available and key details (e.g. proof sketches, experimental setup) are comprehensively described for competent researchers to confidently and easily reproduce the main results.

**Q3 Main Strengths:**

1. A sound and scalable algorithm with theoretical guarantees for data con constrained domains
2. The authored explored the convergence of their approach theoretically
3. The 2d experiments are very nice in illustrating the flexibility of the domains that the algorithm can handle
4. High dimensional experiments showcase the approach works in high dimensions and slightly improves over the unconstrained version (prior work)

**Q4 Main Weakness:**

My main criticism would be that the image-related experiments done by the authors are not necessarily a domain where one can see the benefit of their algorithm, and thus why we see little improvements only when compared to the unconstrained approach. My suggestion to the author would be to look at more interesting data that is supported on compact domains e.g. maybe geopositional data, this would strongly aid the story / strengthen the experimental side.

**Q5 Detailed Comments To The Authors:**

Theorem 2 is pouring into the next column, break the equation into 2 lines for the next version of the manuscript.

**Q9 Complying With Reviewing Instructions:**

Yes

---

> ### Author Rebuttal · Authors · 2024-04-06
>
> We appreciate the valuable comments and suggestions.
>
> **Geopositional data.**
>
> We conducted a brief review of generative models for geopositional data, including recent works [1,2,3,4]. These methods can be broadly categorized into "image generation" (matrix-type sample generation) with pixelated maps or spatial point process generation with continuous-value coordinates. For example, in [1], the matrix represents the map of Japan, where each pixel indicates the temperature at a specific location (see Figure 2 of the paper).  In [4], each sample is the coordination in the map, a two-entry vector of latitude and longitude. As the number of samples is irregular, and the coordination needs to have continuous value, the spatial point process is suitable for this type of generative task. In these cases, the map is usually regular or for a specific region as fixed.
>
> It's worth noting that our method targets the generative task of *regular* vectors, denoted as $x \in \mathbb{R}^k$, where the vector size is fixed. The constrained domain refers to the domain of each component of $x$, such as $x_i \in [0, 1]$. This setup is commonly observed in image generation tasks, where the image has a fixed shape, and RGB values are typically normalized to fall within the range of 0 to 1. This is conceptually different from fixed maps or bounded maps for specific regions.
>
> We welcome the reviewer's feedback on whether they agree with the aforementioned observations or if they have any additional suggestions regarding specific datasets.
>
>
> **Theorem 2 is pouring into the next column, break the equation into 2 lines**
>
> Thank you for your suggestions on the writing and experiments; we will incorporate the changes accordingly in the version.
>
>
> [1] Oyama, N., Ishizaki, N. N., Koide, S., and Yoshida, H. (2023). Deep generative model super-resolves spatially correlated multiregional climate data. Scientific Reports, 13(1), 5992.
>
> [2] Klemmer, K., Koshiyama, A., and Flennerhag, S. (2019). Augmenting correlation structures in spatial data using deep generative models. arXiv preprint arXiv:1905.09796.
>
> [3] Xiao, X., Ganguli, S., and Pandey, V. (2021). Conditional Generation of Synthetic Geospatial Images from Pixel-level and Feature-level Inputs. arXiv preprint arXiv:2109.05201.
>
> [4] González, J. A., Rodríguez-Cortés, F. J., Cronie, O., and Mateu, J. (2016). Spatio-temporal point process statistics: a review. Spatial Statistics, 18, 505-544.

---

### Official Review · Reviewer_usuT · 2024-03-21

**Q2-1 Originality-Novelty:** 3
**Q2-2 Correctness-Technical Quality:** 3
**Q2-5 Clarity Of Writing:** 3

**Q1 Summary And Contributions:**

The paper presents the 'Reflected Schrödinger Bridge', a method for generative modeling for bounded data built on entropy-regularized optimal transport. The method builds on the recently introduced "Reflected diffusion model" [1] and
"Schrödinger Bridge - Forward Backward Stochastic Differential Equations" [2]. The method of the paper demonstrates good performance on several experimental tasks.

[1] Lou and Ermon, 2023, Reflected Diffusion Models, https://arxiv.org/abs/2304.04740
[2] Chen et al, 2022, Likelihood Training of Schrödinger Bridge using Forward-Backward SDEs Theory, https://arxiv.org/abs/2110.11291

**Q2-3 Extent To Which Claims Are Supported By Evidence:**

3: Good: the main claims are supported by convincing evidence (in the form of adequate experimental evaluation, proofs, (pseudo-)code, references, assumptions).

**Q2-4 Reproducibility:**

3: Good: key resources (e.g. proofs, code, data) are available and key details (e.g. proofs, experimental setup) are sufficiently well-described for competent researchers to confidently reproduce the main results.

**Q3 Main Strengths:**

- The method is to my knowledge novel and should pose a valuable contribution to the community.
- The paper is well written.
- The experimental section is in my opinion convincing.
- The paper aims to make generative modelling for bounded domains more rigorous (instead of ad-hoc approaches found often in the literature).

**Q4 Main Weakness:**

- In my opinion, the paper does not really discuss the benefits of using their method over [1] (not w.r.t. performance, but why would I use this over [1] in general) ?
- The submission does not contain experimental code.

[1] Lou and Ermon, 2023, Reflected Diffusion Models, https://arxiv.org/abs/2304.04740

**Q5 Detailed Comments To The Authors:**

- Why did the authors use different generative models for the CIFAR and ImageNet experiments?
- The authors should submit experimental code. Experimental results without code are not reproducible.

**Q9 Complying With Reviewing Instructions:**

Yes

---

> ### Author Rebuttal · Authors · 2024-04-06
>
> We appreciate the valuable comments and suggestions.
>
> **Reflected Schrödinger bridge (rSB) v.s. reflected diffusions and when we prefer rSB**
>
> * $\uparrow$: rSB supports more flexible priors, as shown in Figures 1 and 4, in contrast to reflected diffusion models, which rely on a uniform prior.
>
> * $\uparrow$: rSB supports more adaptable boundaries, as shown in Figures 1, 2, 4, and 11, while reflected diffusion models require the extra stick-breaking procedure to map a simplex domain to a cube, which lacks generality.
>
> * $\uparrow$: rSB provides clear theoretical guarantees on optimal transport, while reflected diffusion models lack such properties.
>
> * $\downarrow$: The use of implicit training loss via rSB introduces a large variance in Hutchinson's estimator, potentially affecting scalability; rSB may not be the best choice for expensive tasks.
>
> We appreciate the insightful suggestions and will include these discussions in the revision.
>
>
> **Why did the authors use different generative models for the CIFAR and ImageNet experiments?**
>
> Schrödinger bridge diffusion models constitute a significant category of diffusion models supporting optimal transport (OT) and are widely applied in generative tasks within image datasets [1, 2, 3]. However, the bounded pixel values in images necessitate ad-hoc thresholding techniques for boundary enforcement, lacking elegance and theoretical guarantees.
>
> To tackle this, the reflected Schrödinger Bridge (rSB) introduces a principled approach for generating a broad range of bounded data. This is achieved through reflected forward-backward stochastic differential equations (SDEs), seamlessly adapting to flexible priors and smooth domains.
>
> [1] Likelihood Training of Schrödinger Bridge using Forward-Backward SDEs Theory.
>
> [2] Diffusion Schrödinger Bridge with Applications to Score-Based Generative Modeling.
>
> [3] Diffusion Schrödinger Bridge Matching.
>
> **Experimental code**
>
> To help replicate the experiments, we have made the experimental code available at https://drive.google.com/open?id=1rAc-EUxTqpp6zb8i3w5idyBLDwwvfpqX&usp=drive_copy. We will release all the code in the next revision.

---

### Official Review · Reviewer_Hco8 · 2024-03-23

**Q2-1 Originality-Novelty:** 3
**Q2-2 Correctness-Technical Quality:** 3
**Q2-5 Clarity Of Writing:** 4

**Q1 Summary And Contributions:**

- This paper studies the problem of learning Schrodinger bridges on constrained domains. This approach generalizes the recently introduced reflected diffusion model to the Schrodinger bridge setting.
- The authors derive a reflected forwards-backwards SDE (Theorem 1), as well as a likelihood-based method for training such SDEs (Prop. 1) with connections to the IPF algorithm.
- Section 4 provides a convergence analysis of the proposed schemes.
- Section 5 includes several illustrations of the approach on 2D toy datasets, as well as some low-dimensional image datasets with competitive but not SOTA results.

**Q2-3 Extent To Which Claims Are Supported By Evidence:**

4: Excellent: all claims are supported by very convincing evidence (in the form of comprehensive experimental evaluation, rigorous mathematical proofs, detailed (pseudo-)code, precise references, well-motivated and realistic assumptions) and the authors deliver what they promise.

**Q2-4 Reproducibility:**

3: Good: key resources (e.g. proofs, code, data) are available and key details (e.g. proofs, experimental setup) are sufficiently well-described for competent researchers to confidently reproduce the main results.

**Q3 Main Strengths:**

- The major strength of this submission are its theoretical results -- learning stochastic processes on constrained domains is a highly non-trivial task. The proposed method is a significant generalization of existing techniques (namely, reflected diffusion models and Euclidean Schrodinger bridges).
- Beyond developing the necessary FB-SDE theory for constrained Schrodinger bridges, the paper also offers a convergence analysis.
- Despite the technical depth of the paper, it is a joy to read and quite clear.

**Q4 Main Weakness:**

- The main weakness of this paper is in its empirical results. The methods are tested on either simple 2D datasets (although, with nontrivial boundaries) or low-dimensional image datasets (CIFAR-10, ImageNet 64x64, MNIST). The proposed method (Table 1) obtains competitive but not state-of-the-art results on these image benchmarks.

**Q5 Detailed Comments To The Authors:**

- The appendix contains many interesting details regarding the actual implementation of the model (Appendix D). I'd encourage the authors  to try to fit some of these details in the main paper, as the implementation appears non-trivial.
- Similarly, I think that placing the related work in the Appendix is not optimal, even though the authors are likely short on space in the main paper (although this admittedly may be a personal preference, and so this point is not considered in determining my score).

**Q9 Complying With Reviewing Instructions:**

Yes

---

> ### Author Rebuttal · Authors · 2024-04-06
>
> *Some important results and related works can be moved to the main draft*
>
> We acknowledge the importance of model implementation and related works on the Schrödinger bridge and reflected diffusion to enhance comprehension. In the next revision, we are committed to enhancing the structure of our work by incorporating these elements into the main draft.
>
> *Empirical experiments*
>
> In addition to the 2D and image datasets, we also conducted generative tasks on simplex domains. As shown in Figure 11, our approach exhibited superior performance compared to the stick-breaking method employed in reflected diffusion models by Lou (2023). We will include more discussions on the experiments in the revision.
>
> Thank you sincerely for taking the time to review our work.

---

### Meta-Review · Area_Chair_9Qs3 · 2024-04-16

A solid and clear consensus by the reviewers recommending accept for a paper that proposes an interesting approach to generative modelling on bounded domains. Please update the paper with the promised changes when revising the paper.